# Resilient Antarctic monsoonal climate prevented ice growth during the Eocene.

Michiel Baatsen[1], Peter Bijl[2], Anna von der Heydt[1,3], Appy Sluijs[2], and Henk Dijkstra[1,3]

[1]Institute for Marine and Atmospheric Research Utrecht, Department of Physics, Utrecht University,Princetonplein 5, 3584CC Utrecht, Netherlands.
[2]Department of Earth Sciences, Utrecht University,Princetonlaan 8a, 3584 CB Utrecht, the Netherlands.
[3]Centre for Complex Systems Studies, Utrecht University, Utrecht, The Netherlands.

**Correspondence:** Michiel Baatsen (m.l.j.baatsen@uu.nl)

**Abstract.** Understanding the extreme greenhouse of the Eocene (56–34 Ma ago) is key to anticipate potential future conditions. While providing an end member towards a distant high emission scenario, the Eocene climate also challenges the different tools at hand to reconstruct such conditions. Besides remaining uncertainty regarding the conditions under which the large-scale glaciation of Antarctica took place, there is poor understanding of how most of the continent remained ice-free

throughout the Eocene across a wide range of global temperatures. Seemingly contradictory indications of ice and thriving vegetation complicate efforts to explain the Antarctic Eocene climate. We use global climate model simulations to show that extreme seasonality mostly limited ice growth, mainly through high summer temperatures. Without ice sheets, much of the Antarctic continent had monsoonal conditions. Perennially mild and wet conditions along Antarctic coastlines are consistent with vegetation reconstructions, while extreme seasonality over the continental interior promoted intense weathering shown

in proxy records. The results can thus explain the coexistence of warm and wet conditions in some regions, with small ice caps forming near the coast. The resilience of the climate regimes seen in these simulations agrees with the longevity of warm Antarctic conditions during the Eocene, but also challenges our view on glacial inception.

## 1 Introduction

Southern Ocean sediment records have revealed perennially warm and wet conditions along Antarctica's continental margins

during the early Eocene, followed by pronounced cooling and amplification of seasonality during the middle and late Eocene (Pross et al., 2012; Contreras et al., 2013, 2014; Passchier et al., 2017; Bijl et al., 2021). In between cool and dry winters, Antarctic summers still had high near surface air temperatures (SATs; >20°C) and precipitation (Robert and Kennett, 1997; Dutton et al., 2002; Basak and Martin, 2013), which seem difficult to accord with reconstructions of partial glaciations on Antarctica (Scher et al., 2014; Passchier et al., 2017; Carter et al., 2017).

Previous modelling studies on the glaciation at the Eocene-Oligocene Transition (EOT) (Gasson et al., 2014; Kennedy-Asser et al., 2020; Sauermilch et al., 2021) have shown a strong sensitivity of Antarctica's climatic conditions to model geography. Large differences between climate models suggest that resolving the regional circulation patterns on and around Antarctica is crucial to simulate a realistic late Eocene climate. Reproducing southern high latitude sea surface temperatures (SSTs) similar

to the proxy record has proven difficult in most modelling studies (Cramwinckel et al., 2018; Kennedy-Asser et al., 2020;
Hutchinson et al., 2021). Climate reconstructions using a poorly resolved Antarctic topography result in an ice sheet growing
primarily from the central highlands, covering most of East Antarctica before reaching the coast (DeConto and Pollard, 2003;
Gasson et al., 2014). Especially precipitation amounts over the continental interior can change drastically in response to altered
(i.e. mainly warmer) near-coastal waters and/or Antarctic topography, which in turn will greatly affect the conditions for ice
growth. Considerable improvement is seen in more recent simulations (Lunt et al., 2017; Hutchinson et al., 2018; Baatsen
et al., 2020; Lunt et al., 2021) using adequate horizontal resolution in the atmosphere ($\sim$2°) and a more elevated Antarctic
paleotopography (Wilson et al., 2012) (see also Figure 1).

Here, we use a set of middle-to-late Eocene (i.e. Lutetian-Bartonian-Priabonian; 48–34 Ma) simulations (Baatsen et al., 2020)
to study the Antarctic climate, prior to glaciation, in more detail. These simulations agree well with a global compilation
of marine and terrestrial temperature proxy reconstructions and cover the range of temperatures observed during this time
interval. Looking at the Antarctic continent in more detail, we aim to explain how Antarctica remained mostly ice free under
considerable temperature swings. We also explore the conditions that allow for the coexistence of regional ice caps and dense
vegetation near the coast, as suggested by the available proxy record.

## 2  Methods

### 2.1  CESM simulations

Our primary results rely on the set of fully-coupled simulations presented by Baatsen et al. (2020), using the Community
Earth System Model version 1.0.5 (CESM1.0.5) with a horizontal resolution of 2.5°$\times$1.9° and $\sim$1°$\times$0.5° for the atmosphere
and ocean, respectively. These simulations consist of: **1)** a hot Eocene case (4$\times$ PIC), **2)** a warm Eocene case (2$\times$ PIC), **3)** a
pre-industrial reference, and **4)** a pre-industrial instant 4$\times$CO$_2$ perturbation. Here, PIC stands for pre-industrial carbon, being
280ppm CO$_2$ and 671ppb CH$_4$, respectively. The radiative forcing of the 4$\times$ PIC and 2$\times$ PIC cases is equivalent to that of
4.85$\times$ and 2.25$\times$ pre-industrial CO$_2$. The Eocene simulations use a 38Ma paleomag-based geography reconstruction (Baatsen
et al., 2016), with shallow marine passages at Southern Ocean gateways. A projected and interpolated version of the Antarctic
topography is shown in Figure 1 (also see Figures S1,S2 in the supplementary material), indicating the main geographic fea-
tures considered in this study. Starting from a homogeneous, motionless state, all model runs have a long spin-up and are well
equilibrated (i.e. 3000–4500 years, globally averaged ocean temperature trends of $\sim$10$^{-4}$K yr$^{-1}$ or less). A more extensive
overview of the model set-up and spin-up procedure for the different simulations is presented in Baatsen et al. (2020). Using
both SAT and SST proxies, the hot 4$\times$ PIC climate is shown to be a good analogue for the late-middle Eocene (42–38 Ma) and
representative for the warmth of the middle Eocene climatic optimum (MECO). With the cooler late Eocene climate being well
represented by the results of the warm 2$\times$ PIC case, these simulations thus capture the range of climate regimes seen within
the middle and late Eocene (Baatsen et al., 2020).

For this work, the existing set of simulations is extended to assess the potential influence of greenhouse gas concentrations,
orbital configurations, and palaeogeography. Besides 4$\times$ and 2$\times$ PIC, we carry out Eocene simulations using pre-industrial

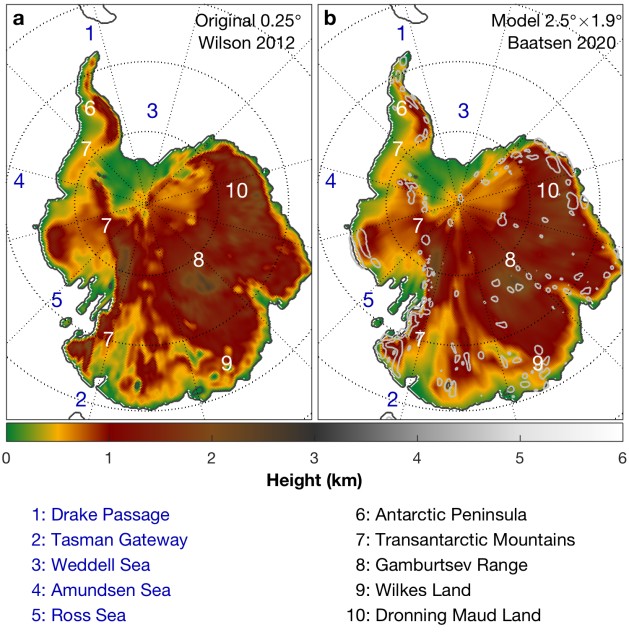

**Figure 1. Projected 38Ma Antarctic topography.**

Antarctic topography reconstruction at 38Ma, projected onto a rectangular $0.25°$ grid centered on the southern pole. **a)** Original reconstruction (Wilson et al., 2012), **b)** Reduced $2.5 \times 1.9°$ version used in the CESM simulations of (Baatsen et al., 2020). Numbers denote oceanic (blue) and topographic (white/black) features of importance, grey contours indicate where the elevation difference between both versions exceeds 250 m.

($1\times$ PIC) greenhouse gas levels. In addition, we consider an orbital configuration with minimum southern high latitude summer insolation, rather than the low eccentricity one used in previous work. Finally, we adopt a 30 Ma-based palaeogeography by Baatsen et al. (2016), albeit with a similar land cover to the 38 Ma reconstruction, including the absence of land-based ice on Antarctica. These '30 Ma Eocene' cases thus still represent Eocene conditions, while using a later palaeogeography reconstruction and serve as an end member for any related uncertainty.

Considering all of the different options would require 12 different Eocene simulations; 3 PIC levels, 2 orbital configurations, and 2 palaeogeographies. An overview of all these possible cases is provided in Table 1. As a trade-off between model complexity, integration length and the available computation resources, we choose to carry out 7 of the 12 possible Eocene simulations (a–g) in addition to 3 pre-industrial simulations (h–j; $1\times$, $2\times$, and $4\times$ $CO_2$). Due to its potential relevance for Antarctic glaciation, the low insolation orbital configuration is only applied to the $1\times$ and $2\times$ PIC cases (b, d, f). The 30 Ma $2\times$ PIC case (d) is started from the same initial conditions as the 38 Ma $2\times$ PIC one (b) and run for 3500 years. The 30 Ma $1\times$ PIC case (f) is branched off from the $2\times$ PIC equivalent (d) and continued for another 3000 years. All simulations with a low insolation orbital configuration (c, e, g) are branched off from the respective existing cases with similar palaeogeography and PIC level, and run for an additional 500 years.

| | Model geography and orbital configuration | | | | |
|---|---|---|---|---|---|
| | 38Ma | 38Ma LS | 30Ma | 30Ma LS | Pre-industrial |
| $4\times$ PIC | **a) E4** | E4L | O4 | O4L | **j) P4** |
| | 4500 years | $c + (a - b)$ | $d + (d - f)$ | $e + (e - g)$ | $h + 2000$ years |
| $2\times$ PIC | **b) E2** | **c) E2L** | **d) O2** | **e) O2L** | **i) P2** |
| | 3500 years | $b + 500$ years | 3500 years | $d + 500$ years | $h + 1000$ years |
| $1\times$ PIC | E1 | E1L | **f) O1** | **g) O1L** | **h) P1** |
| | $b - (a - b)$ | $c - (a - b)$ | $d + 3000$ years | $f + 500$ years | 3000 years |

**Table 1. Model simulations.**

Overview of model cases considered, including equilibrated model simulations (**a–j**; bold) and and extrapolated cases. Each case is given a short name, referring to the model geography (**a–c**) E: 38Ma Eocene, **d–g**) O: 30Ma Oligocene, **h–j**) P: Pre-industrial), level of atmospheric carbon relative to pre-industrial $CO_2$ and $CH_4$ (PIC), and orbital configuration (L for orbit with minimum southern high latitude summer insolation). Below the short names, the number of simulated model years is given as well as the case from which the simulation is started, if applicable. For the remaining extrapolated cases, the respective model output used in the calculation is shown.

Climatologies over the last 100 years of each simulation are used for the analyses and are available in public data repositories. Temperature and precipitation results for the 5 missing Eocene cases are estimated using a linear interpolation of the available data. The respective model simulations used in each of these extrapolations are provided in Table 1. In doing so, we make the underlying assumption that the temperature and precipitation responses to a change in orbital configuration and greenhouse
gas concentrations are both linear and consistent across the different cases. This is indeed supported by the overview figures (S8–S12) provided in the supplementary material. While these interpolated cases can provide further insights into the possible climate regimes of the middle and late Eocene on Antarctica, our primary results depend solely on the available model simulations (a–h).

## 2.2 Climate indices

Next to observable variables, we introduce three climatic indices for a qualitative assessment of the Antarctic climate regimes. The resulting maps and statistics provide an easy visual comparison between all of the different simulated cases. An overview of these indices, applied to our pre-industrial reference, is provided in figures S3 and S4 of the supplementary material. We introduce the following indices, each of which is restricted to the $[0, 1]$-interval:

1. **Glacial index**: $GI = 1 + (1/10) \cdot SMB$, represents the conditions needed to grow land-based ice, using the surface mass
balance ($SMB$, in m year$^{-1}$). We estimate the latter from the total annual precipitation ($P_{ANN}$, in mm) and monthly climatology of SAT. We acquire the positive degree days (PDD) by simply multiplying any monthly average SAT above zero with the number of days for every month of the year. We then use: $SMB = 10^{-3} \cdot (4 \cdot P_{ANN} - PDD)$, assuming a surface melt of 4mm for every positive degree day, similar to e.g. Gasson et al. (2014); Scher et al. (2014). Although this is a very simple estimate of $SMB$, ignoring e.g. the fraction of frozen precipitation and daily temperature variations,

it provides a good indication of whether the climatic regime could be suitable for any kind of ice growth. With the assumptions made here, the surface mass balance is quite optimistic, meaning that it will be difficult to grow any ice if $SMB < 0$ m year$-1$ and extremely unlikely (even with horizontal ice flows) if $SMB < $ -2 m year$-1$. Still, the glacial index will cover a range of SMB between -10 and 0 m year$-1$ to cover a vast range of options under such warm climatic conditions.

2. **Evergreen vegetation index**: $VI = \sqrt{VI1 \cdot VI2}$, where $VI1 = 1/20 \cdot (T_{JJA} + 15)$, and $VI2 = 10^{-3} \cdot P_{ANN}$. The threshold values are based on the parameters associated with temperate and paratropical biomes in proxy reconstructions (Pross et al., 2012; Contreras et al., 2013) This index represents the potential to sustain perennially vegetated conditions, using average austral winter SAT ($T_{JJA}$, in °C) and $P_{ANN}$. $VI$ will approach 1 when the average temperature stays above freezing in winter and annual precipitation is around 1000 mm. To avoid overcompensation between the components, $VI1$ and $VI2$ are each limited to $[0, 1]$ before determining $VI$.

3. **Monsoonal index**: $MI = \sqrt{MI1 \cdot MI2}$, where $MI1 = 2 \cdot P_{DJF}/P_{ANN}$, and $MI2 = 1/500 \cdot P_{DJF}$, based on some of the metrics used by Huber and Caballero (2011). This index represents the degree to which the precipitation is monsoonal in nature, using the average total summer precipitation ($P_{DJF}$, in mm) and $P_{ANN}$. This index mainly considers the fraction of annual precipitation falling in summer, with an additional term to reduce the obtained value when conditions get too dry. $MI$ will approach 1 when summer precipitation represents half the annual precipitation and the latter is about 500 mm. Similar to $VI$, the components $MI1$ and $MI2$ are limited to $[0, 1]$ before determining $MI$.

Before calculating the climate indices, the model output fields are spatially interpolated onto a rectangular 0.25° grid centred on the south pole, using a simple 2D linear scheme. The interpolation is also applied to both the model grid and the original Antarctic topography reconstruction by Wilson et al. (2012) (see Figure 1). All monthly SAT fields from the model are then corrected for the difference in topography, using an 8°C km$^{-1}$ lapse rate. The latter is based on the upper estimate used by Gasson et al. (2014), as it is shown to provide the most conducive regime for ice growth within reasonable limits. In addition to our own simulations, the climate indices are also determined for the early Eocene model simulations of the DeepMIP (Lunt et al., 2017) and presented in Figure S5 of the supplementary material. Despite their simplicity, the climate indices proposed here are able to capture the main climatic regimes of the current climate well. Some ambiguities occur in the tropics where both the monsoonal and evergreen indices are near saturation. In addition, parts of North America and Siberia show relatively high values of the monsoonal index as they are characterised by strong temperature seasonality and wet wet summers.

## 3 Results

### 3.1 Extreme Antarctic seasonality

Under the absence of a continental ice sheet (Coxall et al., 2005; Lear et al., 2008; Scher et al., 2011) and with mostly blocked Southern Ocean gateways (Bijl et al., 2013; Sijp et al., 2014, 2016; Baatsen et al., 2016; Sauermilch et al., 2021), the

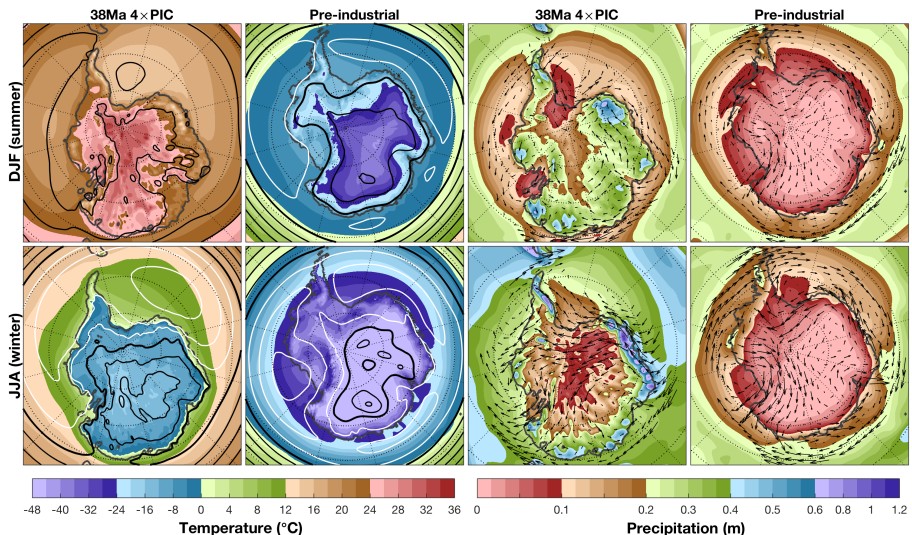

**Figure 2. Seasonal Antarctic surface climate.**

Simulated Antarctic surface conditions in the Eocene 4× PIC case and pre-industrial reference, for the austral summer and winter. Shading indicates seasonally averaged SAT and precipitation, contours show mean sea level pressure (drawn every 5 hPa; black: ≥990 hPa, white: <990 hPa, thick black line at 990 hPa), arrows indicate lower tropospheric flow (100hPa average, only shown over the continent and near-coastal region).

Antarctic climate of the Eocene is drastically different from that of today (Figure 2). In the 4× PIC case the austral summer is characterised by very high SATs over the continental interior. Average summertime temperatures over Antarctica range mostly between 20°C near the coast to over 30°C further inland. Only on higher terrain, cooler temperatures of 10–16°C are found. Winter temperatures stay near or above freezing near the coast, but quickly drop below -10°C moving inland. This means that much of the continent experiences extreme temperature seasonality in these Eocene simulations, with a 40–50 °C difference between average winter and summer conditions. In stark contrast, in the pre-industrial situation on Antarctica, we see cold and dry conditions prevail over the continent even in austral summer. Southern high latitudes are shielded by a belt of cyclonic winds, associated with a strong meridional pressure and temperature gradient. The lowest pressure is found near the coast, rising again over the continental interior due to sinking, cold air. A similar pattern persists during the Eocene winter, albeit with a much weaker belt of steep pressure gradients which is pushed equatorward. The Eocene summer is characterised by overall very weak pressure gradients and thermal low pressure over the continental interior. Few parts of Antarctica receive more than 100mm of precipitation in either the summer or winter in the pre-industrial reference, while most regions do in the Eocene simulations. In both Eocene and pre-industrial cases wintertime precipitation is mostly focused on coastal regions, strongly influenced by the combination of predominant westerlies and the respective Antarctic topography (see Figure 1). An exception to this pattern is found over the Antarctic Peninsula in the Eocene case, where we see most precipitation on the eastern side of the Transantarctic Mountains. The combination of Antarctica's geographical location and absence of sea-ice

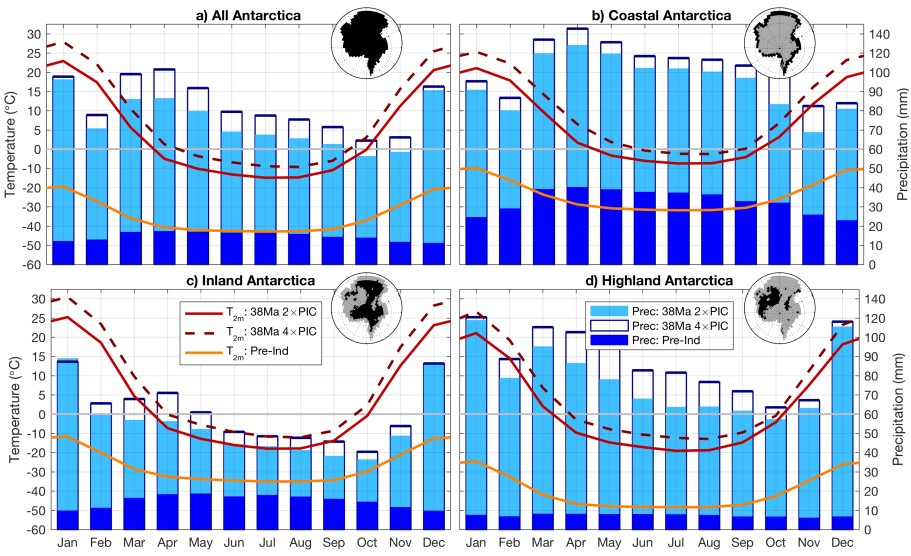

**Figure 3. Spatially averaged antarctic climate.**

Monthly climatologies of SAT (lines) and precipitation (bars), averaged over **a)** the Antarctic continent, **b)** coastal (1 grid cell), **c)** inland, and **d)** elevated regions (>1250 m). The different regions are defined as mutually exclusive and shown as black shading in the map insets.

promote a bipolar pressure pattern, with low pressure located near and over West Antarctica. This pattern results in enhanced onshore flow and precipitation over Dronning Maud Land, as well as easterly winds across the Antarctic Peninsula.

### 3.2 Antarctic summer monsoons

Apart from relatively mild conditions near the coast, most of the Antarctic climate is dominated by strong temperature seasonality. The monthly Antarctic seasonality is presented in Figure 3, showing the average conditions over all, coastal, inland, and elevated (i.e. >1250m elevation) regions. In terms of both SAT and precipitation, there are only small qualitative differences between the Eocene 4× and 2× PIC simulations, with the latter being generally cooler and dryer. Unsurprisingly, the differences between the Eocene cases and pre-industrial reference are much more extreme. The latter is characterised by overall

cold and dry conditions over the entire continent, with the highest precipitation rates occurring in the wintertime and near the coast. Both SAT and precipitation show less seasonal variation compared to the Eocene cases, but are also qualitatively similar between the different regions. While temperature seasonality is generally increased further inland, seasonal precipitation patterns are quite different in the Eocene simulations. Perennially wet conditions are seen near the coast, with a similar seasonal cycle compared to the pre-industrial reference. A peak in precipitation occurs in autumn, starting in March and slowly

declining through the winter and spring. This suggests that precipitation near the coast is mostly a result of baroclinic activity, with a ramp up in activity in autumn followed by prolonged onshore winds during the wintertime. A distinct peak in summer precipitation is seen over the continental interior in the Eocene simulations. While autumn and winter precipitation rates decrease further inland, especially over low-lying regions, the summer peak becomes more pronounced. The rather short, yet

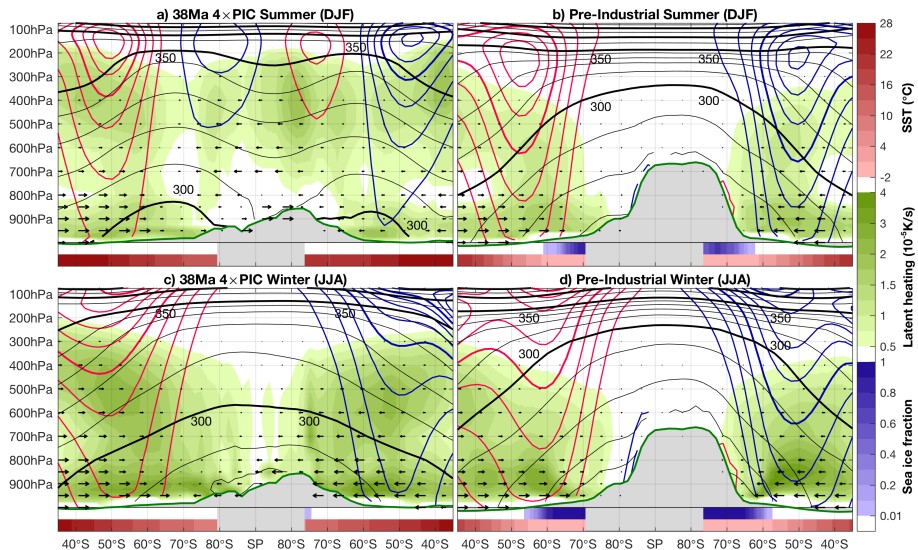

**Figure 4. Components of the Antarctic monsoon.**

Zonally averaged cross section through (left to right) western and eastern Antarctica, for the Eocene 4× PIC case (left) and pre-industrial reference (right), showing Austral summer (top) and winter (bottom). Black contours show potential temperature at 10K intervals up to 350K, thick lines at 50K. Coloured contours show zonal wind every 5 m s$^{-1}$ with thick lines every 20 m s$^{-1}$; blue: into page, red: out of page. Green shading shows the seasonal mean latent heating, arrows indicate the meridional moisture flux. Blue and red colouring is used for sea ice fraction and SST, respectively, with grey shading indicating topographic boundaries. Gray shading indicates the land boundaries, with the grey contour showing the zonally averaged surface pressure.

very warm and wet summer season in the continental interior of Antarctica therefore shows many of the characteristics of a typical sub-tropical summer monsoon.

The different seasonal circulation patterns between the Eocene and pre-industrial simulations are well captured in the zonally averaged vertical cross sections shown in Figure 4. Rather than showing a cross section along a single longitude, we take the zonal average over western and eastern Antarctica. These domains are divided by the 15°W/165°E meridians rather than 0°/180° (see also Figure S1 in the supplementary material), to better distinguish the different terrain types. In the ocean we take the Drake Passage and Tasman Gateway as boundaries, assigning the Pacific sector of the Southern Ocean to western and the Atlantic-Indian sectors to eastern Antarctica.

The altered thermal structure of the atmosphere between the Eocene and pre-industrial simulations is immediately clear by looking at the lines of equal potential temperature (isentropes). In both Eocene and pre-industrial cases, a dome of cold air sits over Antarctica in winter, in which all of the potentially cold air is effectively trapped under adiabatic exchange (following Hoskins, 1991). This strongly limits the exchange of air masses between high and low latitudes and promotes cooling of the polar region. The dome of cold air is much larger and persists through the summer season in the pre-industrial reference. The associated steep thermal gradients at ∼40–70°S demand strong cyclonic winds surrounding the continent. Katabatic flows

emerging from the ice sheet are also seen as an anticyclonic circulation right next to the steepest slopes. These conditions promote year-round baroclinic activity and an associated poleward moisture flux. Persistent storm tracks are evident in plumes of latent heating, caused by forced ascent along the warm conveyor belt of extra-tropical cyclones (Browning, 2004).

The winter circulation pattern in the Eocene is qualitatively similar, but exhibits overall weaker thermal gradients and weaker zonal flow compared to the pre-industrial reference. This would suggest less baroclinic activity, but the associated latent heating is clearly enhanced. Moreover, the moisture fluxes and latent heating reach much further poleward in the simulated Eocene climate. The cross-continental flow shown in Figure 2 can be seen here as well, carrying moisture across Antarctica. The Eocene summer is characterised by a completely different circulation pattern compared to the typical polar winter conditions. Thermal gradients reverse poleward of $\sim60°$S, which through thermal wind balance demand a reversal of the upper level zonal winds. A cyclonic sub-tropical jet is still present at $\sim50°$S, being pushed equatorward and more confined to higher altitude with respect to the pre-industrial reference. Over the Antarctic continent, we see the appearance of a weak anticyclonic summer vortex in accordance to the meridional temperature gradient (i.e. isentropes descending over the pole). The thermal structure of the atmosphere thus shows that the air mass over much of Antarctica is similar to that of the sub-tropics, with the 300–310 K layer crossing the surface in both regions. Onshore moisture fluxes reach the continental interior and are associated with mid-upper tropospheric latent heat release. These are the result of deep convection at high southern latitudes during the Antarctic summer monsoon season.

### 3.3 Regional variation of the Antarctic climate: vegetation and ice

The polar cross section shown in Figure 5 is a good indication of the sharp regional contrasts in the Eocene climate over Antarctica, including the different climatic indices (see Materials & Methods section). Mild temperatures and high precipitation amounts near the coast would allow for the growth of temperate and/or sub-tropical forests. These highly supportive conditions for plant growth are confined to a rather narrow stretch near the coast. A rapid increase in seasonality as well as overall cooling with elevation quickly makes winters too cold for most vegetation to survive. With elevation, we see a sharp increase in precipitation, which is probably underestimated due to the limited model resolution used here. Despite summer temperatures being well above zero, some very high precipitation amounts would allow ice to grow over these regions, especially during cooler intervals. Such ice caps would be strongly restricted to the highest elevations, as well as near the coast, where both temperatures and precipitation are conducive to ice growth. Especially Dronning Maud Land and the Antarctic Peninsula are good candidates for the formation of ice caps in our simulated Eocene climate, Nevertheless, the proximity of such ice caps to the coast, in combination with cool upper ocean temperatures and westward currents, could allow for the calving of ice bergs floating along the Antarctic Peninsula. Further inland, we see a quick change towards monsoonal conditions, which dominate much of the Antarctic continent. While temperature seasonality is highest over the continental interior, precipitation overall is also lower thus decreasing the (still largest) monsoonal index. The stark regional differences in the simulated Eocene climate are thus mostly determined by the distance to the coast and by the topography.

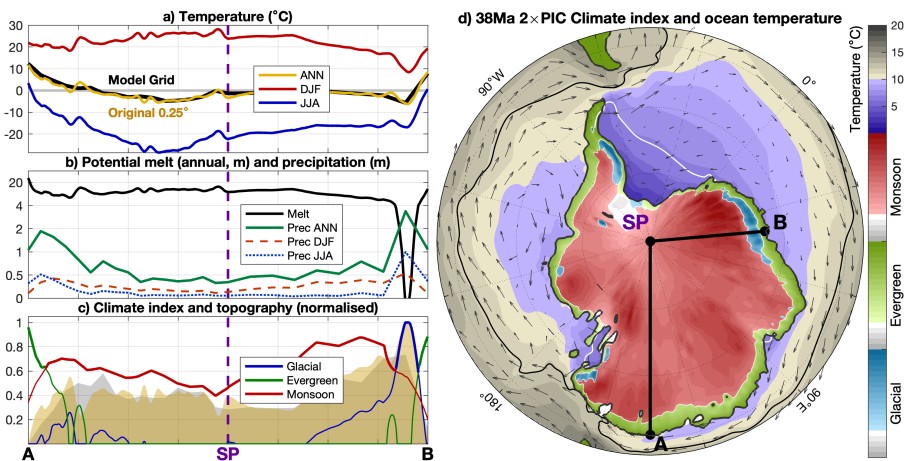

**Figure 5. Antarctic climate regimes.**

Antarctic meridional cross section from Wilkes Land (A), through the south pole (SP), to Dronning Maud Land (B) for the Eocene 2× PIC case. Profiles are drawn for **a)** SAT, **b)** potential melt and precipitation, and **c)** climate indices and topography (grey: model, yellow: full resolution). **d)** Trajectory of the cross section (black), spatial pattern of the largest climate index (shading), and annual mean ocean temperature averaged over the upper 200m (blue-gray shading, top part of colour bar, contours for the 4× PIC case; white: 10°C, black: 15°C). Arrows represent the upper 100m average flow, the maximum arrow length corresponds to 2 cm s$^{-1}$.

### 3.4 Resilience of the Antarctic climate under possible Eocene conditions

Up to this point, we have only considered the results of the standard 38Ma cases and the pre-industrial reference. To check the robustness of the Antarctic climatic regimes shown in Figure 5d, we look at their distribution over the Antarctic continent for 12 different Eocene and 3 pre-industrial scenarios (Figure 6). The same climate indices are used, but using 2 simple criteria rather than a continuous scale for easy comparison between the different scenarios. We consider surface mass balance (at -2 and 0 m year$^{-1}$) for the glacial index, winter SAT (at -5 and 0 °C) for the evergreen, and summer precipitation (at 100 and 300 mm) for the monsoonal one. A figure showing the standard climate indices, as well as a histogram showing the spatial contribution of each index can be found in the supplement material (Figures S6 and S7, respectively). The supplement also provides specific overviews of SAT (Figures S8, S9), precipitation (Figures S10–S12) and surface mass balance (Figure S13) between the different scenarios.

Besides an overall cooling trend, the result of lowering atmospheric greenhouse gases on the vegetation index is not straightforward. Between the 4× and 2× PIC cases, wintertime cooling and drying mostly reduce the extent of the evergreen vegetation regime towards the coastline. Meanwhile, further enhanced seasonality slightly enhances the monsoonal index. We see the appearance of glacial conditions at 2× PIC, although very limited in spatial extent. Towards 1× PIC, there is a much more prominent growth of the glacial regime at the expense of both the vegetation and monsoonal ones. The cooler scenarios mostly limit the vegetation and monsoonal regimes, rather than allowing substantial glaciation. An orbital configuration with low summer insolation over southern high latitudes acts to reduce seasonality, mainly in temperature. This reduces potential summer

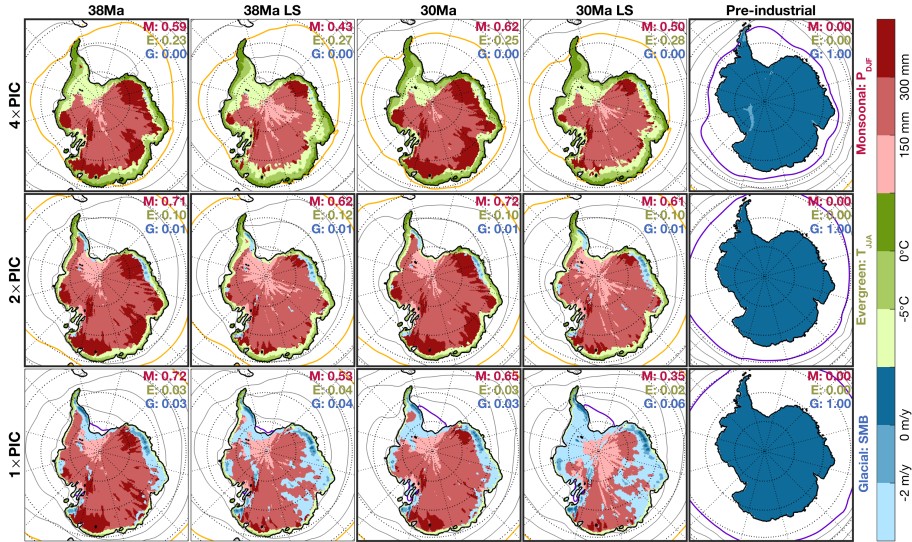

**Figure 6. Resilience of Antarctic climate regimes.**

Overview of climate index regimes between the different simulated (thick boxes) and extrapolated cases. LS indicates cases with low summer insolation orbit (see also Table 1). Glacial index is subdivided by surface mass balance, evergreen vegetation by winter SAT, and monsoonal index by summer precipitation. Numbers show the fractional coverage across the Antarctic continent of each index, for which the first criterion is met (e.g. SMB>-2 m year−1 for glacial).

melt of ice, but also reduces overall precipitation. At 4× and 2× PIC, the low summer insolation cases therefore reduce the monsoonal regime while increasing the evergreen vegetation. At 1× PIC, cooler temperatures reduce the vegetation regime while improving glacial conditions. The effect of altering the paleogeographic reconstruction between 38Ma and 30Ma (i.e.

mainly widening and deepening Southern Ocean Gateways) has a limited effect on the Antarctic climatic regimes. A slight cooling and drying of the continent again induce a small increase of the glacial regime, especially in the 1× PIC cases. Despite being the dominant climate index over a substantial area, the glacial regime is still mostly characterised by a strongly negative surface mass balance. Over much of the continental interior, summertime warmth and low precipitation still greatly limits the potential for ice growth, which is consequently limited to near-coastal elevated regions through the Eocene. In spite of consid-

erable regional SST changes between both reconstructions, similar to those shown by Sauermilch et al. (2021), much of the Antarctic continent thus sees only a minor influence in terms of possible glacial conditions.

### 3.5   Comparison to available proxies

A global overview of available proxies of terrestrial and marine temperatures for the middle-to-late Eocene, compared to model simulations, is provided by Baatsen et al. (2020). Here, we focus on a comparison between proxies and our 38Ma 4× PIC and

2× PIC case for the Antarctic region in Figure 7, including annual SAT, seasonal SAT, annual precipitation and annual upper 200m ocean temperature. Land-based proxy records are considered from Seymour Island (annual SAT: Dutton et al. (2002),

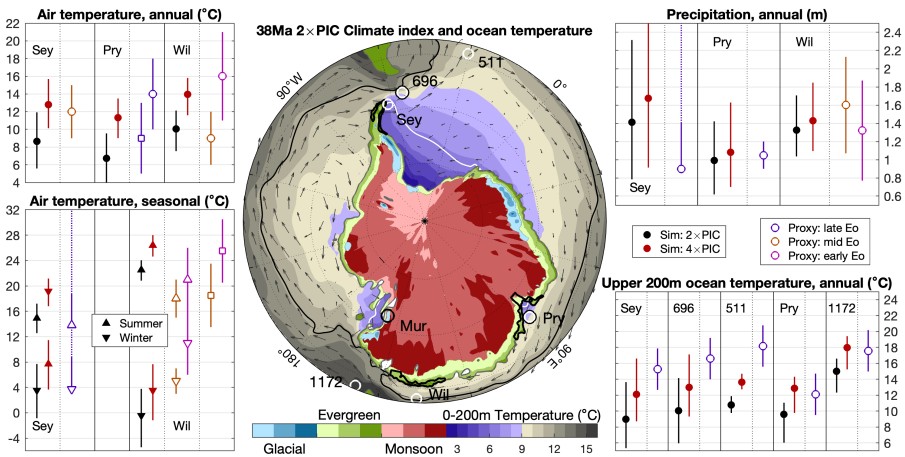

**Figure 7. Comparison to the proxy record.**

Climate index regimes over Antarctica for the 38Ma 2× PIC case and upper 200m ocean temperature, similar to Figure 5d, using a slightly adjusted scale and the simplified indices as in Figure 6. Relevant proxy sites to this study are indicated using white circles; Sey: Seymour Island, Pry: Prydz Bay, Wil: Wilkes Land Margin (U1356), Mur: McMurdo; numbers indicating ODP/DSDP sites. Black circles indicate sites at which ice rafted debris has been reported, pre-dating the EOT. Surrounding panels provide a comparison between available proxies (open markers) and corresponding model values (solid markers) for annual/seasonal SAT, annual precipitation and annual upper 200m ocean temperature. Uncertainty margins for the model data indicate spatial variation, while reported error margins on the different proxy sources are used (with dashed lines indicating lower bounds).

seasonal SAT and annual precipitation: Mörs et al. (2020)), Prydz Bay (annual SAT and precipitation: Tibbett et al. (2021)), and the Wilkes Land Margin (U1356; Pross et al. (2012); Contreras et al. (2013)). Terrestrial proxies are compared to model values, considering a region surrounding the 38Ma location of each site by 10° longitude and 5° latitude. Only data on the

235 Antarctic continent is considered at an elevation of at most 300m to exclude mountainous terrain. The area-weighted average, minimum and maximum of the remaining datapoints are then used as best estimate and uncertainty bounds of the model data. Ocean temperatures are considered from Seymour Island (Douglas et al., 2014), ODP 696/South Orkney (Hoem et al., 2023), DSDP 511/Falkland Plateau (Liu et al., 2009), Prydz Bay (Tibbett et al., 2021), ODP 1172/East Tasman Plateau (Bijl et al., 2013, 2021). Here, the upper 200m ocean temperatures are estimated from reported TEX86 values using the calibration of Kim

et al. (2012). Corresponding model temperatures and uncertainty bounds are obtained within a 4° radius around the paleolocation of each proxy site, averaged over the upper 200m (i.e. 20 model levels).

The fossil frog record from Seymour Island only provides a lower estimate of coldest/warmest month SAT and annual precipitation. The Wilkes Land Margin record only provides early and middle Eocene (∼54–46 Ma) estimates, covering annual/seasonal SAT and annual precipitation using sporomorph assemblages, and MBT/CBT which is noted to be representative of summer

temperatures. There is an overall good agreement between the model results and the different proxy records. Despite the considerably earlier dating, the different Wilkes Land proxies correspond well with our simulations, although the temperature

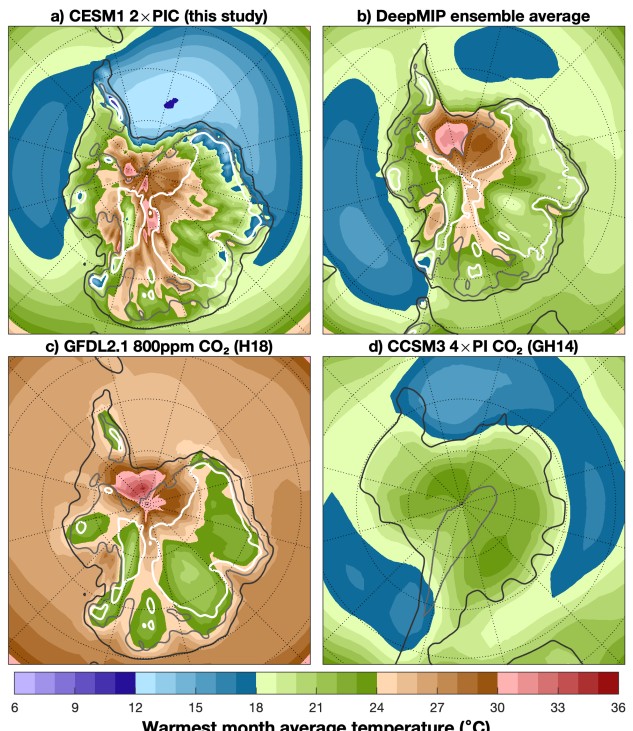

**Figure 8. Comparison between Eocene model studies.**

Warmest month average SAT over southern high latitudes in Eocene simulations from different studies: **a)** our 38Ma 2× PIC case, **b)** the multi-model mean 3× PI $CO_2$ early Eocene simulatins from DeepMIP (Lunt et al., 2021), **c)** 38Ma 800ppm $CO_2$ GFDL CM2.1 simulations from Hutchinson et al. (2018), and **d)** middle Eocene 4× PI $CO_2$ simulations from Goldner et al. (2014). Dark contours show the coastline based on model land fraction, gray and white contours show the model topography at 500m and 100m, respectively.

seasonality is still stronger in the model. Ocean temperatures show a more mixed agreement, with model temperatures close to the proxy estimates on the Indo-Pacific/Tethyan side but rather cool on the Atlantic side. Although mostly within the uncertainty range, modelled ocean temperatures are consistently below the proxy estimates at Seymour Island, ODP 696 and DSDP 511 even when considering the 4× PIC case. In addition, the latitudinal gradient in ocean temperature seems to be underestimated in the model compared to the available proxies. That said, temperatures near the Antarctic coastline agree well with the different temperature estimates provided by the proxy record. This shows that the Antarctic climate is indeed captured well by our 38Ma 4× PIC for the middle Eocene and 38Ma 2× PIC for the late Eocene, respectively.

### 3.6 Comparison to existing modelling work

An overview of the warmest month average SAT from different modelling studies is shown in Figure 8. Similarly warm summertime conditions on Antarctica are seen between our study, early Eocene simulations from DeepMIP (Lunt et al., 2017, 2021), and the 800ppm GFDL simulations from Hutchinson et al. (2018). The latter use the same paleogeography from

Baatsen et al. (2016) as our 38Ma cases, but have a slightly lower horizontal model resolution. Earlier work using CCSM3 by Goldner et al. (2014) does not show these warm temperatures over the Antarctic continent. Meanwhile, SSTs are quite similar between DeepMIP, CCSM3 and our study while they are much warmer in GFDL.

Although there are considerable differences among the models, the DeepMIP ensemble shows an overall pattern of the climate indices consistent with our study (see Figure S5 of the supplementary material). A monsoonal climate regime dominates most of the continental interior of Antarctica, while vegetation thrives along the continental fringes. There is less consistency regarding the potential glacial conditions, especially regarding the spatial distribution, but this may be the result of the topography as well as the lacking lapse rate correction. Still, there is an overall agreement that conditions on Antarctica would be hostile to grow substantial volumes of land ice during the Eocene, i.e. very few areas show a surface mass balance of $> -2$ m year$-1$. Most of these scenarios would only allow for the existence of local ice caps, while most of the continent remains either monsoonal or vegetated. Limited availability of the different model topographies does not allow us to implement a similar temperature adjustment as in our own results (used in Figure 6). While this may underestimate some potential for glacial conditions, this is restricted to highly localised elevations, as shown by the comparison in Figure 1.

## 4   Discussion and conclusions

### 4.1   Simulated Eocene Antarctic climate conditions

In our simulations the Antarctic climate of the middle-to-late Eocene is characterised by high seasonality and warm, wet summers. Coastal regions experience mild and wet winters, with a sharp transition towards much colder winters further inland. High latitude warmth is possible under the absence of a continental-scale ice sheet, aided by warm SSTs and a lacking deep circumpolar current. Intense summer warmth over the Antarctic continent reverses the meridional temperature gradient and breaks down the otherwise dominant cyclonic polar vortex. The release of latent heat in deep convection adds to the warming over Antarctica and enhances the poleward flow of moisture. Most of the continent sees mild mean annual SATs of 2–12 °C and 400–1700 mm of precipitation. On average, about half of the annual precipitation falls in summer over inland regions. Much of the continental interior sees a climatic regime similar to present-day sub-tropical monsoons, but also share characteristics with wet summer regimes in parts of North America and eastern Siberia. Active storm tracks reaching high southern latitudes boost autumn and winter precipitation, aided by topographic lift which is particularly strong along the East Antarctic coast. Most of the coastal regions see a coldest month SAT near or above freezing (kept warm by ∼10°C waters). A 50–60 °C SAT difference is seen between the mean coldest and warmest month over central Antarctica, values only seen over parts of Siberia in the present climate. Considering warm and wet summers as well, eastern Siberia may thus provide the closest present analogue to the Eocene climate over much of Antarctica. The extreme seasonality seen in our simulations puts some question marks regarding the boundary conditions used, imposing a cool mixed forest over most of Antarctica for the Eocene cases. Our results would suggest a more lush vegetation type near the coast on one hand, and a more barren type over the continental interior. While some further testing (not shown here) does indicate that applying a tundra vegetation over most of Antarctica

influences SAT, the effect is minor (localised changes of 1°C or less) and does not alter the main results. If anything, a more barren vegetation type would act to further increase temperature seasonality over inland Antarctica in the Eocene.

## 4.2 Agreement with available proxies

The conditions seen on Antarctica in these model simulations generally fit well with vegetation reconstructions for the middle and late Eocene, with the warmer scenarios being representative for the early Eocene as well. Our model results show mild and perennially wet conditions in Antarctic coastal regions, which would support a cool-temperate (Nothofagus) forest or sub-tropical vegetation. This is in agreement with available temperature and precipitation reconstructions for the middle Eocene Wilkes Land east Antarctic Margin (Pross et al., 2012; Contreras et al., 2013), Seymour Island (Dutton et al., 2002; Douglas et al., 2014; Mörs et al., 2020), and Prydz Bay (Tibbett et al., 2021). The regional occurrence of paratropical vegetation in the early-middle Eocene is supported by our 4× Eocene cases. Frost weary vegetation could easily survive in coastal regions even under relatively modest radiative forcing. Mild, ever wet conditions allow for the presence of sub-tropical rain forests on a significant part of Antarctica, extending beyond the continental fringes during warmer intervals. Wet temperate conditions are reconstructed for several near-Antarctic coastal sites prior to the Eocene-Oligocene boundary (Amoo et al., 2022; Thompson et al., 2021), suggesting that these warm and wet conditions indeed prevailed until the onset of continental-scale Antarctic glaciation. Chemical weathering indicates a climate with warm and wet summers, while physical weathering and various fossil records support high seasonality (Scher et al., 2011; Basak and Martin, 2013). In combination with summer rainfall, this seasonality suggests the presence of a summer monsoon as suggested by Jacques et al. (2014).

## 4.3 Robustness of Antarctic conditions

We tested the consistency of the Eocene climatic regimes in our simulations, considering the influence of atmospheric greenhouse gases, the paleogeographic reconstruction, and the orbital configuration. Consistent with earlier work, the climatic regimes are most sensitive to a reduction of greenhouse gas concentrations (Gasson et al., 2014; Goldner et al., 2014; Anagnostou et al., 2016; Kennedy-Asser et al., 2020), while the other effects are relatively minor. A cooling of Antarctica is, however, also related to a reduction of precipitation. Even in the optimal scenario for ice growth explored here, most of the Antarctic continent is still a hostile place for large-scale glaciation. The remarkable resilience and suggested reversibility of the Antarctic monsoonal climate presented here could thus explain well why the continent resisted glaciation for many million years, regardless of the occurrence of cold intervals with regional ice caps.

## 4.4 Consistency with other modelling studies

Some previous model studies found a relatively warm and wet climate on Antarctica during the early Eocene (Huber and Caballero, 2011; Huber and Goldner, 2011). Seasonality and annual rainfall are less pronounced in those simulations, likely related to the representation of the Antarctic continent. The combination of a considerably lower Antarctic palaeogeography reconstruction and limited model resolution results in the CCSM3 simulations missing most of the Antarctic summer warmth

as well as the sharp regional differences seen in our work. In this sense, the simulations presented here are among the first to discuss a regionally variable climate on Antarctica in the Eocene with a sufficiently resolved continental geometry. The result is a more extreme, warmer and wetter Antarctic climate that exhibits several characteristics of a summer monsoon. Our findings are consistent with a modelling study using the GFDL climate model (Hutchinson et al., 2018), as well as more recent modelling efforts within DeepMIP (Lunt et al., 2021). While showing similarly strong seasonal variation on Antarctica during the Eocene, neither of these studies consider the Antarctic climate in particular. Despite the focus on early Eocene conditions, all of the model contributions to the DeepMIP reproduce the summer warmth on Antarctica seen in our simulations, being mostly comparable in terms of climatic regimes. These results clearly suggest that primarily the palaeogeography, but also the model version/resolution, determine our ability to accurately represent the Antarctic Eocene climate.

### 4.5 Consequences for ice growth

Apart from some isolated regions (Dronning Maud Land, Antarctic Peninsula and Transantarctic Mountains), ice sheet growth is highly unlikely even at relatively low radiative forcing (below the $\sim$2.5–3 $\times$ $CO_2$ threshold suggested in earlier studies (DeConto and Pollard, 2003; DeConto et al., 2008; Gasson et al., 2014)). Notably, these isolated regions lie very close to the coast and favour ice growth due to a combination of high precipitation (up to 4m annually) and cool temperatures (at >2km elevation), which agrees well with the suggested prevalence of mountain glaciers in the Eocene (Barr et al., 2022). The position of the Antarctic continent is of importance, as it is shifted from the South Pole with respect to today (van Hinsbergen et al., 2015; Baatsen et al., 2016). As a result, the winter circulation splits into a high/low configuration over East and West Antarctica, respectively (Figure 3). The cross-continental flow in between acts to enhance winter precipitation over both Dronning Maud Land and the Antarctic Peninsula. Surface ocean currents in the Weddell Gyre (see Figure 5d) allow the transport of ice rafted debris from marine terminating glaciers at both these locations towards where they are found at ODP Site 696 (Carter et al., 2017) (South Orkney Microcontinent).

### 4.6 Main findings and conclusions

The simulated Antarctic climate presented here is characterised by large regional differences. Most of the continent experiences extreme seasonality and is characterised by a subtropical-like monsoonal climate. There is good agreement with the limited available proxy record in terms of temperature and precipitation estimates. These climatic features appear to be particularly resilient across a wide range of possible conditions throughout the Eocene. The model results presented here are in good agreement with recent modelling work, which shares the differences with respect to earlier studies: an updated Antarctic topography and increased model resolution.

While indications of ice on Antarctica (Scher et al., 2014; Passchier et al., 2017; Carter et al., 2017; Barr et al., 2022) are seemingly in disagreement with warm and wet conditions (Pross et al., 2012; Contreras et al., 2013; Tibbett et al., 2021; Thompson et al., 2021; Amoo et al., 2022), the model results presented here are able to reconcile both features. Thriving vegetation near the coast can likely persist through much of the Eocene, migrating further inland or back towards the coastal fringes along with long-term fluctuations in temperature. Near-coastal regions with higher elevation see a combination of high

precipitation, cool summers and cold winters which permits the growth of glaciers and small ice caps. Especially during cooler intervals in the late Eocene, these ice caps could grow considerably. Summer warmth in most of the continental interior of Antarctica still prevents the further growth of localised ice sheets, indicating that considerable regional climatic changes are needed before a continental-scale Antarctic ice sheet can form at the end of the Eocene.

*Data availability.* All the model output is post-procesed using PYTHON 2.7.9 and MATLAB. Maps using a polar stereographic projection are generated using the M_MAP package, available at www.eoas.ubc.ca/~rich/map.html. A selection of the model data used to generate the main figures in this paper are publicly available on the Utrecht University Yoda platform;

  – 38Ma 4× PIC Eocene CESM simulation: https://doi.org/10.24416/UU01-UFU2KD

  – 38Ma 2× PIC Eocene CESM simulation: https://doi.org/10.24416/UU01-A9JXH1

  – Pre-industrial reference CESM simulation: https://doi.org/10.24416/UU01-KHITZQ

The above data is post-processed to be more accessible and only contains the variables considered in this specific work. The full data from the respective model simulations are available upon reasonable request from the authors.

*Author contributions.* MB conceived the idea for this study, after which all authors contributed to the conceptualisation of the narrative and analyses needed. MB, AvdH, and HD designed the model simulations. MB post-processed the data, conducted the analyses, and constructed the figures. PKB, AS, and MB assembled the proxy data. All authors contributed to the writing of the manuscript.

*Competing interests.* Some authors are members of the editorial board of CP. The peer-review process was guided by an independent editor, and the authors have also no other competing interests to declare.

*Acknowledgements.* The authors thank Michael Kliphuis for assisting with the model simulations and management of the output data. We thank Simon Michel for his help with the additional simulations conducted for this study. We also want to thank Edward Gasson and David Hutchinson for their help with DeepMIP comparison as well as open discussions.

This work was carried out under under the program of the Netherlands Earth System Science Centre (NESSC), financially supported by the Ministry of Education, Culture and Science (OCW, grant #024.002.001). Simulations were performed at the SURFsara dutch national computing facilities and were sponsored by NWO-EW (Netherlands Organisation for Scientific Research, Exact Sciences) under the projects 17189 and 2020.022. This work has further received support from the TiPES project, funded by the European Union's Horizon 2020 research and innovation programme under grant agreement No 820970 (TiPES contribution #213).

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
