# Peer review of "Resilient Antarctic monsoonal climate prevented ice growth during the Eocene."

_Climate of the Past, 2023_

## Author Response (AR1)

**Overview (reviewer 1)**

This paper focuses on Antarctic climate during the extreme hothouse world of the Eocene (defined here as 56-34 million years ago), with the aim of better anticipating future climate changes. A state-of-the-art global climate model is used to reproduce Eocene climate under various scenarios of $CO_2$, and the paper concludes that extreme seasonality in Antarctic climate limits ice growth, instead allowing most of the continent to have monsoonal conditions. The results also show the resilience of the various Antarctic climate regimes.

The paper is, for the most part, well written, with a clear structure including an excellent abstract (nicely summarising the main findings), followed by an introduction, brief methodology, results and discussion section. Rather than just considering basic climatic variables, the authors use three indices - a glacial index, an evergreen vegetation index and a monsoonal index - to assess their simulations, thereby bringing a novel approach to this area of research. Combined with the state-of-the-art GCM simulations, this paper is therefore both novel and timely. My recommendation, therefore, is to accept this manuscript, subject to some minor revisions as outlined below.

**Major comments**

My main comment is that currently a section on model-data comparison is completely missing and is needed. Model-data comparison is mentioned briefly at the very end of the introduction, but is a single sentence only. It is mentioned again in slightly more detail at the end, in Section 4.2, but this focuses more on vegetation reconstructions. Before acceptance, I would like to see another section, perhaps immediately before the model results are presented, giving a more in-depth analysis of how key variables (e.g. temperature and precipitation) from the various simulations match the available proxy data. At the moment this is missing, which makes it harder to determine the accuracy/reliability of these simulations and therefore the conclusions.

*AR: As our main focus in the original manuscript was to provide a more qualitative interpretation of the Antarctic climate, we chose not to include a specific section on model-proxy comparison. Instead, we referred to a more elaborate comparison in the previous paper discussing the model results and have a more specific discussion of Antarctic proxies further down. Following the main remarks of both reviewers, we added a section on a more quantitative mode-proxy comparison to the results section. This makes the results more complete and less reliant on earlier publications.*

Another, less major, comment would be that some restructuring is needed. For example, Section 3.5 appears to overlap with Section 4.4; I would I would recommend moving 3.5 down into the discussion and merging with 4.4. Likewise Section 3.4 appears to discuss the same matter as Section 4.3, and therefore could be merged into the latter discussion section. Likewise Section 4.1 appears to be a summary of the main findings, which appears to overlap with Section 4.6 and again could be merged into one section at the end.

*AR: There is indeed some overlap between the results and discussion sections, which can be reduced by some restructuring. Moving 3.5 into discussion, however, would imply moving Figure 7 into the discussion therefore presenting results (at least partly). Most of the results and discussion section were reconsidered and restructured or adjusted where needed to reduce. This is helped further by adding a results subsection on the model-proxy comparison.*

**Minor comments**

- Abstract: I would like to see a little more motivation - the authors say that understanding Antarctic climate during the Eocene is "*key to anticipate potential future conditions*", but why?

  *AR: Fully ice-free conditions on Antarctica are off course far away from any expected near-future scenario, but are still indicative of what can be expected in terms of high latitude warmth in a more general sense. We added some more motivation/nuance on this part.*

- Results: As discussed above, this needs a new section containing a model-data comparison.

  *AR: we agree that this can help to support the model results and included an additional subsection with the new figure 7 in the results section.*

- Throughout the manuscript: brackets (or some sort of other punctuation) are needed around references e.g. from the first paragraph, it currently reads "… *during the middle and late Eocene Pross et al. (2012); Contreras et al. (2013, 2014); Passchier et al. (2017); Bijl et al. (2021)*" whereas something like "… *during the middle and late Eocene (Pross et al. 2012; Contreras et al. 2013, 2014; Passchier et al. 2017; Bijl et al. 2021)*" would be better.

  *AR: Thank you for pointing this out, we adjusted the references where needed.*

- Figures: I find some of these quite hard to read/interpret, either because they use very bright and alternating, fairly garish colours (e.g. Figure 2) or they are quite small (e.g. Figure 6). Could these be re-drawn to aid interpretation?

  *AR: these figures were adjusted and redrawn to improve on the different aspects mentioned.*

**Reviewer 2**

The manuscript authored by Baatsen et al. delves into a comprehensive exploration of the climate conditions prevalent in ice-free Antarctica during the Late Eocene and Early Oligocene epochs. The paper accentuates the pivotal role played by seasonality in shaping the climate of this ancient Antarctic landscape. While the prevailing temperature and precipitation patterns appear conducive to the proliferation of temperate or subtropical flora along the coastal regions, aligning well with empirical observations, an intriguing contrast emerges when examining the conditions further inland. Inland areas seem to present a paradoxical scenario: too cold during the winters to permit the growth of vegetation and yet too warm during the summers to facilitate glacial inception, unless we consider the presence of formidable high mountain ranges. The paper is notably commendable for its clarity and well-crafted prose. Its structure is clear.

However, despite its strengths, I do have a few comments and suggestions to offer."

The formatting of bibliographic references in the manuscript appears to be incorrect and requires correction.

*AR: thank you for pointing this out, we corrected the reference formatting.*

L25: "a different representation of coastal waters": please clarify this sentence.

*AR: This refers to the large differences in near-coastal currents and temperatures compared to most previous simulations, we adjusted this to clarify.*

L46: A sentence explaining how climate equilibrium is defined is necessary.

*AR: we added some additional explanation on the temperature trends seen in the simulations, referring the reader to earlier work for further details.*

L69 : I'm uncertain about how interpolation can reliably yield accurate results for the five missing Eocene cases. Additionally, there is some ambiguity in Table 1 regarding the notation 'E2+E2-E4.' It's not clear to me what this notation signifies in the context of the integration of simulations or the number of years simulated from the starting point corresponding to the end of a simulation."

*AR: Different parts of information are given in this table, apparently leading to some confusion. On one hand, we show simulation length for the cases that are simulated, including information on how/when they were branched off. On the other hand, we indicate how results are being interpolated from the available simulations. We adjusted the table and added an alphanumeric order to the full simulations to clarify this. Regarding the interpolation: we indeed make a rather strong assumption of full linearity here, which is explained in the methods. Our simple methodology for this specific use is supported well by the large overall consistency between cases shown in the results. Additionally, we do not use any of these extrapolated datasets to make a detailed assessment of the Antarctic climate. This is now motivated and explained in more detail in the methods.*

L80: To my knowledge, SMB is not used in Goldner et al 2014.

*AR: This is indeed the case, we removed the reference here.*

L87: A reference for the evergreen vegetation index is needed

*AR: A reference was added regarding the values used in the index, 2 new supplementary figures were added to show the climate indices for our pre-industrial simulations.*

L93 : Different types of monsoon indices are available, and it's worth noting that this paper's monsoonal index, tested with ERA5 dataset, demonstrates relatively robust performance, particularly at lower latitudes. However, the results show an index close to 1 in regions such as eastern Siberia, the Pacific coast in South America, and North America. This prompts the question of whether this criterion is adequately restrictive.

*AR: In their current form, the indices used are indeed quite simple and therefore not completely restrictive. As suggested above, we added a figure to show this for pre-industrial conditions, along with some motivation and discussion here. We do not claim that the Antarctic Eocene climate is strictly identical to that of present-day sub-tropical monsoons, but simply much alike in several ways. Being at a high latitude, there will always be some fundamental differences regardless of overall temperature and humidity. In fact, the present-day climate of Eastern Siberia and parts of North America may be a rather close analogue to much of Eocene Antarctica, with cold and dry winters being alternated by warm and wet summers. We also added this to the discussion.*

Figure 2 could provide valuable insights into the wind direction, which should undergo a reversal or significant change with seasonal fluctuations in sea level pressure. Yet, understanding changes in atmospheric circulation is somewhat challenging when examining sea level pressure due to the lack of labels on isocontours, especially for the summer at 38 Ma, where thick contours are absent. To enhance clarity and insight, it is advisable to include a plot of low-tropospheric wind and sea level pressure in the Supplementary Information section."

*AR: This is indeed tricky, as MSLP gradients become rather weak overall in the Eocene summer climate on Antarctica. We agree that it would be helpful to add a visible indication of low-tropospheric flow here. Rather than adding a figure, we chose to show quivers over the precipitation fields instead of repeating the MSLP contours.*

L101 : How was the lapse rate fixed ?

*AR: The constant lapse rate of 8C/km is commonly used in the modelling of high-latitude ice sheets. This suits our overall qualitative approach and is motivated by simplicity. We added some more explanation and/or referencing here.*

Figure 4 : Add labels on isocontours.

*AR: We added labels to the thick contours at 300K and 350K, to improve readability.*

Figure 5: in caption, "and" is missing between potential melt and precipitation

*AR: We corrected this.*

L176 and L245: The authors conclude that the prevailing temperature and precipitation conditions are conducive to the growth of temperate forests along the coast but not in the inner continent. It's worth considering whether the vegetation prescribed in the simulations (Baatsen et al. 2020) as a cool/warm mixed forest remains consistent despite the observed seasonality. Another question arises regarding the representation of the inner Antarctica continent. Given the simulations, it appears that a bare soil with a higher albedo might provide a more accurate representation of this region. It can be useful to explore the potential impact of such a change in representation on the overall conclusions of the study.

*AR: Our results indeed put some question marks on the vegetation that was implied in the simulations. Some testing with alternative (tundra-like) vegetation types suggested only minimal impacts, with mainly a slight further in seasonality over the continental interior. We chose not to include these in the results, but added some discussion here.*

L254: The model-data comparison is very brief. This section needs to be completed.

*AR: As our main focus in the original manuscript was to provide a more qualitative interpretation of the Antarctic climate, we chose not to include a specific section on model-proxy comparison. Instead, we referred to a more elaborate comparison in the previous paper discussing the model results and have a more specific discussion of Antarctic proxies further down. Following the main remarks of both reviewers, we now added a subsection and figure showing a more quantitative mode-proxy comparison to the results section. This should make the results more complete and less reliant on earlier publications.*

L269: a "s" is missing in "consistent"

*AR: Thank you for pointing this out, we corrected this.*